# Bacterial Vaginosis and Vulvovaginal Candidiasis Pathophysiologic Interrelationship

**DOI:** 10.3390/microorganisms12010108

**Published:** 2024-01-05

**Authors:** Jack D. Sobel, Yogitha Sai Vempati

**Affiliations:** 1C.S. Mott Center for Growth and Human Development, 275 E. Hancock St, Detroit, MI 48201, USA; 2Department of Internal Medicine, Wayne State University School of Medicine, Detroit, MI 48201, USA; yvempati@med.wayne.edu

**Keywords:** bacterial vaginosis, vulvovaginal candidiasis, fluconazole resistance, recurrent vaginitis, microbiota, microbiome, pathogenesis

## Abstract

Among the infectious causes of vulvovaginal symptoms, bacterial vaginosis (BV) and vulvovaginal candidiasis (VVC) dominate. Apart from infrequent mixed infections, both are considered independent and caused by unrelated pathogenic mechanisms. Clinical experience, however, is strongly suggestive that in some populations these infections are linked with recurrent BV (RBV) serving as the dominant etiopathogenic trigger for development of recurrent VVC (RVVC) with profound clinical and therapeutic consequences. The biologic basis for this critical interrelationship is discussed and suggests that as a consequence of BV dysbiosis, and not necessarily because of antibiotics prescribed, immune defenses are compromised, neutralizing vaginal yeast tolerance. The consequent BV-induced vaginal proinflammatory environment predisposes to mixed infection or consecutive episodes of post-treatment VVC. Recurrent BV and repeated antimicrobial drug exposure also predispose to acquired fluconazole resistance in *C. albicans* isolates, contributing to refractory vulvovaginal candidiasis.

## 1. Introduction

Bacterial vaginosis (BV) and vulvovaginal candidiasis (VVC) are the two most common varieties of vaginal infection. Each presents a unique spectrum of clinical manifestations, and they are viewed as separate entities of markedly different etiopathogenesis and causation. Although infrequent, mixed infections with simultaneous characteristic expression of both BV and VVC do occur. Little is known about linkage and disease interaction. In a study of women with RVVC in Argentina, an association with BV was found in 35% and intermediate vaginal microbiota in an additional 33.2% of women [1]. For clinicians, the only well-recognized link is the development of VVC in the following days or few weeks after a symptomatic episode of BV treated with appropriate antibiotics (Figure 1). A reasonable causal explanation is that antibacterial agents impact the already pathologic microbiome of BV and remove any residual bacterial restraints or “brakes”, facilitating vaginal yeast proliferation, leading to symptomatic VVC. The treatment of the BV episode rather than the BV infection itself is thought to result in the consecutive episode of VVC. This phenomenon is well known to both patients and practitioners, so much so that women with this highly predictable complication of BV treatment routinely request a simultaneous prophylactic oral antifungal drug, usually fluconazole, when receiving antibacterial agents. Prophylactic fluconazole is highly effective in this context, but its use is largely unvalidated. This relationship between BV and VVC has therefore been limited to a drug-induced VVC attack. Little thought or enquiry into other linkages has been explored, nor are data available on the frequency of this occurrence or the clinical and therapeutic implications. BV is widely recognized as predisposing to multiple sexually transmitted infections, but little attention has been directed at VVC as a consequence of BV [2].

As clinicians in a tertiary care vaginitis clinic, we evaluate women referred for management of either recurrent BV (RBV) or RVVC, with more than 80% of women referred for RBV declaring a history of RVVC. Vaginal yeast infections are viewed by patients as the non-dominant problem in terms of personal suffering and only viewed as an unavoidable collateral effect of needed antibiotic therapy for BV with no thoughts by practitioners of an alternate biologic association. In the authors’ experience, dealing with a predominantly African American population served in Detroit, MI, although a variety of unrelated causes and triggers of RVVC are described, we propose that RBV is the dominant contributor to RVVC [3,4].

In order to explore any link or relationships between BV and VVC other than antibacterial therapy for BV, it is necessary to review aspects of the pathogenesis of VVC and BV. 

## 2. Pathogenesis of Vulvovaginal Candidiasis (Abbreviated Summary)

The primary pathogens in VVC are *Candida* species, likely originating from the gastrointestinal tract and achieving initial asymptomatic vaginal colonization as commensals. *Candida* microorganisms enjoy a saprophytic existence in a moderated non-adversarial environment created by vagino-protective microbiota. In particular, organic acids, both acetic and lactic acid, contribute to vaginal tolerance of *Candida* spp. [5]. The presence of a tolerant cytokine atmosphere emphasizes the critical role of the vaginal mucosa and its expressed defensive function [2,3,4]. The local effect of estrogen plays a dominant role in the healthy lower genital tract microbiome in allowing *Candida* commensalism. Vaginal yeast colonization can be long term and is influenced by host genetic influences together with acquired colonizing promoting factors, both behavioral and biologic [3,4]. The outcome of benign asymptomatic colonization as well as *Candida* vaginitis reflects the interplay of three factors: yeast, vaginal microbiota, and host mucosal immune factors. A detailed description of human vaginal mucosa, host immunity, and the role of vaginal microbiota in the pathogenesis of VVC has been reviewed by several investigators [3,4,5,6,7,8,9,10]. A critical role of host local innate immunity, in both defense and pathogenesis of VVC, is widely proposed with resultant symptoms and signs the consequence of host no-longer protective inflammatory response [3,4,5,6,7,8,9,10,11,12,13,14]. 

Acute symptomatic *Candida* vulvovaginitis represents a dramatic change triggered by multiple factors but always requiring prior vaginal yeast colonization and characterized by proliferation of yeast blastospores and hyphae formation with expression of multiple fungal virulence factors, and these microbiome changes result in superficial vaginal epithelial surface invasion and consequent vaginal epithelial cell proinflammatory reaction. The accompanying myriad signs and symptoms evident in acute vulvovaginitis soon follow. Both IL-1β and IL-6 are increased as part of the proinflammatory mucosal response [3,4,7,8]. Risk factors for acute VVC include vaginal dysbiosis after antimicrobials, increased estrogen, and uncontrolled diabetes, all superimposed upon genetic susceptibility consisting largely of single nucleotide polymorphisms [3,15]. All members of the triad contribute to the risk and expression of disease. VVC may be acute, short-lived, chronic, or recurrent. Unlike oral candidiasis, immunosuppression is not a prerequisite for VVC.

We still lack a full understanding of yeast virulence factors, vaginal microbiota, and host mucosal immune mechanisms involved in acute vaginal candidiasis. Even common “triggers” that precipitate transformation from commensal to pathogenic fungal state and vaginitis are poorly understood, preventing further understanding of the links between BV and VVC.

## 3. Pathogenesis of Bacterial Vaginosis (Abbreviated Summary)

Unlike VVC, with a single or mono-pathogen pathogenesis, BV represents a severe polymicrobial vaginal dysbiosis with loss or disappearance of what is considered “healthy” protective *Lactobacillus* species and overgrowth of multiple largely strict anaerobic and facultative species, creating a more diverse bacterial abundance. This includes multiple *G. vaginalis* species, *Fannyhessea vaginae*, *Mobiluncus* spp., and BVAB (BV-associated bacteria 1–3) and *Bacteroides* spp., *Clostridiale* spp., *Prevotella* spp., *Zozaya* spp., and others [16,17]. Whether the overgrowth of these pathogenic species contributes to the reduction or elimination of *Lactobacillus* species or alternatively follows their disappearance has not been conclusively established. Many investigators favor the introduction or emergence of the pathologic consortium of anaerobes as the primary process, possibly related to sexual transmission from a male or female partner [18]. Some virulent *Gardnerella* spp. are thought to be a key factor in BV pathogenesis, displacing vaginal *Lactobacillus* spp. and adhering to vaginal epithelial cells. Considerable evidence indicates the role of sexual transmission to explain multiple recurring BV episodes in women followed longitudinally, although sexual transmission or pathogen reintroduction is clearly not the only mechanism of frequent recurrence or relapses [19]. 

More recently, the recognition and appreciation of the BV biofilm coating the surface of the vaginal mucosa has contributed to understanding the pathogenesis of BV, especially RBV, and has improved treatment of BV. Virulent strains of *Gardnerella* spp. are thought to initiate biofilm production and are often the predominant species present in the biofilm [19]. It is hypothesized that certain *Gardnerella* species act to lower the oxidation reduction potential of the vaginal micro-environment as well as elaborating essential substrates (NH_3_), allowing growth of strict anaerobic bacteria. The microbiome of BV has undergone extensive investigation in the last decade, significantly enhanced by the availability of PCR and next-generation sequencing, leading to advances in diagnosis, but it has not, to date, contributed to treatment advantage [20,21,22,23]. A conceptual model of BV pathogenesis including initial invasion with *G. vaginalis* and *Prevotella bivia*, followed by a second wave of colonizers including *Atopobium vaginae* and *Sneathia* spp., has been proposed [23].

Turning to the third component of the causal triad, the role of the host immune system in the pathogenesis of BV and expressed in the vaginal mucosa has been extensively investigated. BV has long been considered a “non-inflammatory” condition, hence the term “vaginosis” and not “vaginitis” [24]. This early definition or designation was largely driven by lack of typical clinical signs of vaginal and vulvar inflammation including pain, soreness, or dyspareunia. This decision was reinforced by the striking absence of polymorphonuclear leukocytes (PMNs) in the vaginal exudate or discharge on saline microscopy. However, this assumption did not match later immunologic scrutiny. The vaginal environment in BV is profoundly proinflammatory, as confirmed in multiple studies [20,25,26,27]. Cytokine and chemokine increase are evident with increase in IL-2, type 1 interferon. The lack of PMNs appropriately reflects the effect of chemokine or chemotaxic inhibitors preventing their accumulation. The role of vaginal inflammatory mediators in the pathogenesis of BV is largely unknown and is perhaps crucial to the loss of protective *Lactobacillus* species as well as to explaining clinical BV recurrence and resistance to probiotic therapy. 

## 4. Linking BV and VVC

As mentioned above, not all women with RVVC have associated BV or RBV. In fact, many women with RVVC have no such association. On the other hand, in some women prone to RBV, clinical experience reveals that RVVC is an enormous and predictable additional problem [28]. Invariably, it is BV that precedes and appears to trigger RVVC episodes. The first question to be addressed is whether vaginal *Candida* colonization rates are increased in women with BV. In fact, several vaginal microbiota studies in women with BV reveal increased rates of *Candida* vaginal colonization [28]. Similarly, *Candida* colonization rates in our clinic in women with acute BV, even in the absence of positive microscopy for yeast elements, are approximately 30–35%, significantly higher than in matched women serving as normal controls (12–15%). This is not surprising given the higher vaginal pH characteristic of BV reflecting loss of lactic acid and bacteriocin-producing protective *Lactobacillus* species. This is particularly evident in *L. iners*-dominant communities characteristic of BV, both during acute attacks of symptomatic BV as well as following successful treatment of BV with metronidazole [29,30]. As a general principle, therefore, vaginal dysbiosis facilitates *Candida* colonization [2,6]. 

## 5. Antibiotic Treatment of BV Triggers Consecutive VVC

Antimicrobial use is widely accepted as a common, if not the most common, trigger of acute VVC [31]. Virtually all antibiotics cause this complication, although certain classes of antibiotics carry a greater risk. Remarkably, the mechanisms by which antibiotics facilitate proliferation of commensal yeast are poorly understood. It has been proposed that *Candida* proliferation is the consequence of antibiotics lysing resident bacterial species with release of unknown bacterial components that activate yeast virulence factors. However, little is known of the exact process. An alternative mechanism is that antibiotics, by eliminating “protective” bacterial species, result in reduction of “yeast-inhibiting” factors. The now unrestrained commensal yeast blastopores in turn proliferate, form hyphae, and release virulent enzymes, which facilitate epithelial cell adherence and invasion of the vaginal epithelial cell barrier. All the above is hypothesized to occur in the vaginal lumen but could also occur in the adjacent gastrointestinal tract. Evidence of antibacterial agents directly stimulating commensal yeast proliferation has not been forthcoming, however, antibiotics have recently been shown to impact host local innate immunity and hence, indirectly, anti-mycotic activity [32].

Accordingly, antibiotic therapy for RBV is the most frequently recognized trigger for RVVC, more commonly seen with the lincosamide class, i.e., clindamycin, and less frequently with the nitroimidazole class (i.e., metronidazole). Both topical and systemic routes of administration are responsible, with few published data of VVC superinfection rates associated with drug class or treatment route. In a recent study of maintenance therapy utilizing twice weekly metronidazole vaginal gel for RBV, VVC recurrence was demonstrated in >50% of patients, requiring the addition of maintenance weekly fluconazole to allow continued prophylactic metronidazole for prevention of RBV [33]. 

With regard to the second concept that antibiotics predispose to VVC by reducing or eliminating protecting *Lactobacillus* species, the weakness of this hypothesis is that women with symptomatic BV already demonstrate profound depletion of protective *Lactobacillus* species well before antibiotics are administered. Furthermore, following the administration of clinically effective metronidazole, the post-treatment vaginal microbiota prior to onset of consecutive VVC is still dominated by *Lactobacillus iners* species, which produce less lactic acid and in vitro inhibit *Candida* less effectively than *L. crispatus* [29,30,31]. Similarly, following effective treatment of BV, rapid decline in acetic acid occurs, allowing *Candida* spp. to proliferate [34]. Thus, the predilection to consecutive VVC post BV is complex and likely due to several simultaneous changes in the microbiome including higher pre-antibiotic treatment *Candida* colonization and further loss of protective *Lactobacillus species,* which are replaced by the less protective *L. iners*. 

## 6. Bacterial Vaginosis Microbiota and VVC (Figure 2)

While vaginal dysbiosis predisposes to vaginal *Candida* colonization, the question arises as to the effect of dysbiosis on *Candida* virulence [35]. Firstly, a predictable and consistent feature of BV is an elevated vaginal pH. It is widely acknowledged that pH homeostasis and adaptation to its change are critical for the growth and virulence of *C. albicans* [36]. Acidic pH due to lactic and acetic acid represses yeast-filamentous growth, while neutral and alkaline pH conditions promote filamentation, facilitating yeast colonization and tissue invasion [37,38]. The protection of lactobacilli, now lost in BV, can also be attributed to bacteriocins’ competitive exclusion, enhancing yeast access to nutritional support and access to epithelial cell adherence sites.

**Figure 2 microorganisms-12-00108-f002:**
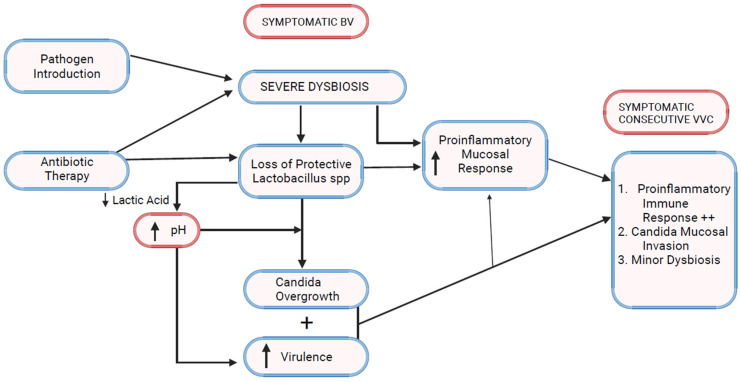
Bacterial vaginosis–vulvovaginal candidiasis pathophysiologic linkage.

A characteristic consequence of anaerobic proliferation in BV is the increased elaboration of biogenic amines including putrescine, cadaverine, and trimethyl amine oxide responsible for the characteristic malodor observed in women with BV. Bacterial polyamines have been shown in vitro to inhibit yeast budding and germ tube formation [39]. This phenomenon has been hypothesized to explain the relative infrequence of mixed infections involving BV and VVC but may explain the delayed consecutive appearance of VVC after successful antibiotic therapy of BV with eventual reduction of polyamines and decline of a yeast virulence inhibitory mechanism. 

While most attention examining vaginal bacterial yeast interaction has focused on the role of *Lactobacillus* spp., such interactions likely occur with other bacterial species that are part of BV dysbiosis [40,41]. In studies reported with *Enterococcus faecalis*, bacteriocins that inhibit hyphal morphogenesis, biofilm formation, and virulence of *Candida albicans* are reported [42]. *Enterococcus* spp. are increased as part of BV dysbiosis at the same time that yeast virulence inhibitory factors derived from lactobacilli are decreased, hence other inhibitory bacterial mechanisms need additional exploration [41]. 

## 7. Bacterial Vaginosis Modulates Host Immune and Inflammatory Responses

*Lactobacillus* spp. possess not only antimicrobial, including antifungal, activity but also immunoregulatory properties [43]. *Lactobacillus* dominance is considered essential to a physiologic anti-inflammatory environment. Importantly, changes in vaginal bacterial composition reciprocally stimulate host transcriptional and immune responses that alter the associated vagino-cervical microbiome [2,44]. A more diverse microbiota is linked to high mucosal inflammation levels and a compromised genital epithelial barrier [26]. Despite the wide prevalence of BV, there remains an unmet need to better understand the immunologic microenvironment accompanying and driving the dysbiotic state and the contribution of the individual BV-associated species to this environment and disease process. *Gardnerella* spp. and *Prevotella* spp. are not strongly proinflammatory, however, the secondary wave of BV microbiota including *Fannyhessea vaginae* and BVAB secondary colonizers (*Sneathia* spp.) enhances the vaginal inflammatory response [23]. One can propose a possible immunologic link between BV and VVC with the high community diversity and dysbiosis characteristic of BV, inducing genital mucosal proinflammatory cytokine concentrations (IL-1, IL-8, IL-16, TNF-α, INF gamma, MCP-1) transforming commensal *Candida* organisms but also reacting to the new fungal threat so inducing an inflammatory vulvovaginitis clinical response [2,20,21,45,46]. As mentioned earlier, antibiotics also directly impact host mucosal innate immunity [32].

With time and early recovery from BV following antibiotic therapy, vaginal pH changes can be expected with recovery of lactic acid-producing species. The progressively acidic environment promotes yeast cell β-glucan surface exposure, leading to further hyperactivation of the host vaginal innate immune response [47,48]. The latter effect adds to the growing mucosal hyper-reactivity directed at the increased number of vaginal yeasts, leading to consecutive symptomatic VVC. 

## 8. Impact of BV–VVC Interaction on Fluconazole Resistance

Recently, attention has been directed to the increased prevalence of fluconazole resistance in *Candida* spp. causing bloodstream infections. Similarly, an increase in fluconazole-resistant *C. albicans* vaginitis is widely reported [49,50,51]. In both situations, past fluconazole exposure is considered a major contributory factor. An additional factor in candidemia patients is thought to be systemic antibacterial therapy exposure. *C. glabrata* bloodstream infection was strongly associated with recent metronidazole exposure [52]. In addition, infection with fluconazole-resistant blood isolates is associated with exposure to beta-lactam antibiotics and clindamycin [52]. Metronidazole and clindamycin, both oral and topical, remain the foundation of treatment of BV with repeated exposure in women with RBV. In a recent review of patients seen in the Vaginitis Clinic at Wayne State University, from whom fluconazole-resistant *C. albicans* isolates were isolated, the frequency of women with RBV was evaluated. During 2021 and 2022, 50 *C. albicans* isolates with fluconazole resistance were identified, and 34 (68%) patients had a history of recurrent BV. An even higher frequency of fluconazole-resistant *C. albicans* isolates was evident in patients with mixed BV–VVC infections. Similar observations were made by Arecharvala et al. [1]. Accordingly, evidence is emerging of yet another complication of RBV in which repeated courses of appropriate antibacterial agents, both 5-nitroimidazoles and clindamycin, not only cause consecutive VVC post BV but specifically fluconazole-resistant VVC due to *C. albicans*. This complication is even more unfortunate when one considers the large number of symptomatic women worldwide receiving inappropriate antibiotics empirically for unconfirmed BV [53]. 

## 9. Discussion

BV is the most prevalent vaginal infection in women worldwide. BV, although more common in women who are immunocompromised, nevertheless predominates in women who are immune competent with normal quantities of immune mediators in the vaginal environment [2]. Accompanying BV development, increased vaginal levels of proinflammatory immune mediators are present, which, together with severe dysbiosis, are considered complicit in poor reproductive health outcomes [2]. These include preterm birth, human immunodeficiency virus acquisition, as well as a variety of sexually transmitted bacterial and viral infections. While VVC is usually included in the list of lower genital tract infections complicating BV, such infections (VVC) are generally considered of little consequence. 

While vaginal dysbiosis following and secondary to antimicrobial therapy undoubtedly accounts for post-BV vaginal candidiasis, a complex picture emerges in which a heightened proinflammatory milieu adds to the pretreatment of existent dysbiosis of BV and creates an ideal storm for both mixed and consecutive VVC infections. The clinical expression of VVC is dependent upon and is the result of an enhanced proinflammatory vaginal mucosal response. In spite of the considerable progress made in understanding the immunopathogenesis of oral candidiasis, the same level of progress has not been witnessed in vulvovaginal candidiasis. Notably missing in VVC is an effective role for neutrophils in providing protection, possibly related to local complement deficiency. The normal non-protective proinflammatory vaginal mucosal response is only effective in the presence of vaginal eubiosis and low numbers of colonizing *Candida* microorganisms representing a purposely down-regulated host innate defense mechanism. However, in the presence of severe dysbiosis, such as is seen in BV and the expanding population of proliferating *Candida*, the consequent non-neutrophil proinflammatory response is not protective but adds to the clinical signs and symptoms characteristic of VVC and RVVC following RBV. 

There is much to learn as to how the vaginal microbiota influence host immune function and vaginal mucosal inflammation and how this modulation impacts disease susceptibility and expression. Multiple studies consistently demonstrate that high diversity of vaginal microbiota correlate strongly with genital proinflammatory cytokine concentration [2,20,23]. In fact, BV can be considered the “poster child” example of a high diversity that lacks *Lactobacillus* dominance. This kind of microbiome not only induces but also enhances genital tract proinflammatory cytokines, which we propose predispose to vulvovaginal candidiasis. The intracellular mechanisms activating the inflammasome and proinflammatory pathways may be non-identical for each pathogenic species. VVC, in its own right, in the absence of BV or dysbiosis, is usually viewed only from the perspective of “yeast pathogen dependent” in its various states as a colonizer, commensal, or vaginal pathogen [54]. VVC reflects a state of heightened proinflammation of incompletely determined causation but is an example of failed host defense mechanisms in which therapeutic antifungals indirectly control inflammation and hence symptomatic disease by reducing mycotic load but do not cure RVVC. Recurrent VVC may be primary or secondary to a variety of host behavioral and/or non-behavioral factors. A well-defined exaggerated monocyte-derived cytokine response to *Candida* hyphae is documented in women with idiopathic RVVC, reflecting an underlying genetic susceptibility [55]. 

The important role of RBV as a dominant contributory factor to VVC recurrence in some subpopulations is underappreciated by practitioners but certainly not by patients. The use of prophylactic fluconazole together with antibiotic therapy for RBV is reasonable in those women with a history of RVVC and has become the norm in regional therapeutic practice. The use of maintenance suppressive once weekly fluconazole is already widely accepted as standard of care in women with idiopathic RVVC [56]. In women with RVVC in whom recurrent BV is the dominant pathogenic driving process, such prophylactic antifungal regimens may be unnecessary, and attention is possibly focused on control of the RBV. The association of RBV with fluconazole resistance is similarly unrecognized and worthy of further study. Undoubtedly, frequent fluconazole exposure, when used for both treatment and prophylactic purposes, is the likely dominant cause of secondary fluconazole resistance of *C. albicans* in these patients. Whether dysbiosis or inflammatory cytokines also contribute to resistance is speculative and requires further investigation. 

## Figures and Tables

**Figure 1 microorganisms-12-00108-f001:**
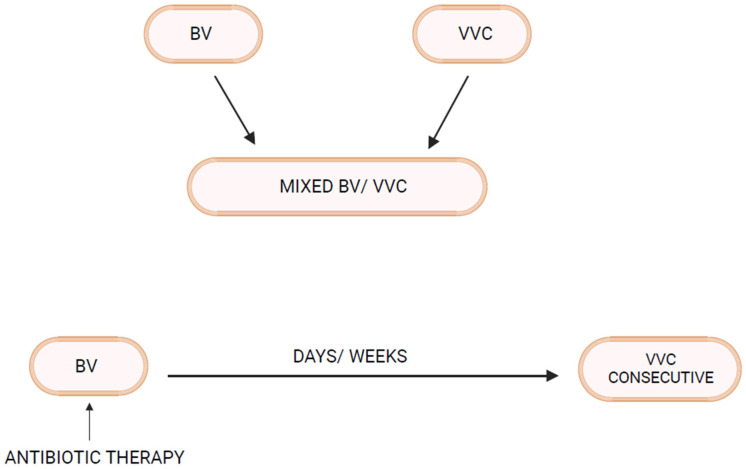
Bacterial Vaginosis–Vulvovaginal Candidiasis interrelationship.

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
