# Peer review of "Bacterial Vaginosis and Vulvovaginal Candidiasis Pathophysiologic Interrelationship"

_microorganisms, 2024, doi:10.3390/microorganisms12010108_

Round 1
Reviewer 1 Report
Comments and Suggestions for Authors
Vulvovaginal candidiasis (VVC) and bacterial vaginosis (BV) are the most common vaginal infections associated with a deep negative impact on women's quality of life. VVC affects about 75 % of all women at least once in their life, and 8 % of them globally develop a recurrent form (RVVC). BV is a severe polymicrobial vaginal dysbiosis with a high global prevalence, ranging from 23% to 29% of women, with significant variation by race and geographic area, and with a high possibility of recurrence. Co-infection between Candida and bacteria is also possible, but this condition is still understudied and difficult to evaluate because of the complex vaginal environment. During a co-infection, the woman's quality of life decreases even further, with mixed symptoms characteristic of both infections. Pharmacological control of VVC, BV, and recurrent forms, while possible with maintenance antifungal-antibacterial therapy, remains problematic and does not eliminate the risk of both future reinfection and/or co-infection and the development of drug resistance.
In this review, the authors describe well the pathogenesis of VVC and BV infections alone and the link between the two major vaginal infections, considering the influences of the microorganism and host and using a large amount of recent and clinical studies. Trying to highlight and demonstrate the link between VVC and BV is crucial to improve therapeutic strategies and avoid the development of drug resistance.
The review is a significant contribution to the field because it is one of the first papers trying to link VVC and BV. Moreover, considering the shortage of studies on this topic, the authors have done an elegant and valuable research work, clearly interconnecting all the available information.
Author Response
Thank you for reviewing our manuscript and providing a commendable response.
Reviewer 2 Report
Comments and Suggestions for Authors
The manuscript „Bacterial Vaginosis and Vulvovaginal Candidiasis Pathophysiologic Interrelationship”, written by Sobel J. et al is a very interesting review, approaching not only the bacterial vaginosis and the vaginal candidosis, but also the interrelationship between them.
The manuscript is well organized, taking step by step the candidosis, bacterial implication and the relation between them. I would recommend the following changes:
- the references are not in the format of the journal, aspect which can be corrected in the last form of the manuscript
- in the subtitles 2 and 3 what do you mean through abbreviated summary?
- please add a figure regarding the actual relation between bacteria and Candida and their action on the vaginal mucosa cells, with all their molecules and with the changes from colonization to infection for Candida.
- reduce/replace some of the self-citations
Author Response
Thank you very much for reviewing our manuscript and we greatly appreciate your inputs.
Comment 1: the references are not in the format of the journal, aspect which can be corrected in the last form of the manuscript.
The references format has been corrected.
Comment 2: in the subtitles 2 and 3 what do you mean through abbreviated summary?
Referring to a brief or short summary.
Comment 3: please add a figure regarding the actual relation between bacteria and Candida and their action on the vaginal mucosa cells, with all their molecules and with the changes from colonization to infection for Candida.
No changes were required.
Comment 4: reduce/replace some of the self-citations.
No changes have been made.